# A Unique Case of Unilateral Pseudogynecomastia

**DOI:** 10.3390/diagnostics14182058

**Published:** 2024-09-17

**Authors:** Ismini Kountouri, Ioannis Katsarelas, Eftychia Kokkali, Amyntas Giotas, Christos Gkogkos, Dimitrios Chatzinas, Panagiotis Nachopoulos, Afroditi Faseki, Alexandra Panagiotou, Athanasios Polychronidis, Miltiadis Chandolias, Nikolaos Gkiatas, Dimitra Manolakaki, Periklis Dimasis

**Affiliations:** 1Department of General Surgery, General Hospital of Katerini, 60132 Pieria, Greece; giannis24katsarelas@gmail.com (I.K.); jimmys055@gmail.com (D.C.); panosnacho@gmail.com (P.N.); afroditifaseki@gmail.com (A.F.); alexandrapanayiotou.larissa98@gmail.com (A.P.); thanospolychronidis@gmail.com (A.P.); miltoshandolias@gmail.com (M.C.); nikgiat71@gmail.com (N.G.); dimanolakaki@gmail.com (D.M.); dimasis@yahoo.com (P.D.); 2Department of Radiology, General Hospital of Katerini, 60132 Pieria, Greece; kokkalieutuxia@gmail.com; 3Gynecology and Obstetrics Department, General Hospital of Katerini, 60132 Pieria, Greece; ammag10@live.com (A.G.); akisgogos@yahoo.gr (C.G.)

**Keywords:** gynecomastia, pseudogynecomastia, occupation-related diseases

## Abstract

Background/Objectives: Gynecomastia is a common condition characterized by the benign enlargement of male breast tissue, often resulting from hormonal imbalances. A rare variant, unilateral pseudogynecomastia, involves enlargement due to adipose tissue accumulation without glandular proliferation and can be associated with occupational factors. Methods: We report the case of a 45-year-old male mechanic presenting with unilateral enlargement of the left breast. The patient reported daily microtrauma on his left axilla and chest wall. The clinical evaluation and imaging revealed lipomatosis with pronounced fibrous tissue and no glandular tissue involvement. The hormonal assays were within the normal limits. The patient underwent surgical excision of excess adipose tissue using the Kornstein technique, preserving the nipple–areola complex. Results: The histopathological examination confirmed the absence of malignancy. The postoperative recovery was uneventful, and the follow-up examination at 12 months demonstrated a symmetrical breast appearance with no recurrence. This case underscores the importance of differentiating pseudogynecomastia from true gynecomastia and recognizing potential occupational risks. Surgical management using techniques that preserve the nipple–areola complex can achieve excellent cosmetic outcomes.

Gynecomastia is a condition characterized by the persistent benign enlargement of mammary gland tissue in men, which can be either unilateral or bilateral [1]. True gynecomastia is usually a result of hormonal fluctuations [1]. In cases of physiologic pubertal gynecomastia, a relative excess of estrogen triggers breast enlargement, which usually regresses with increased testosterone production during late puberty [1]. The standard diagnostic workup for true gynecomastia includes laboratory tests regarding aspects such as human chorionic gonadotropin (hCG), luteinizing hormone (LH), thyroid-stimulating hormone (TSH), testosterone, and estradiol levels [1]. Approximately 2–4% of gynecomastia cases are related to testicular tumors, and 7–11% of patients with testicular tumors present with gynecomastia as the initial symptom [2]. Thus, clinical examination involves palpation of the breast to differentiate between fatty tissue and glandular tissue, as well as testicular palpation to detect early testicular changes or atrophies, guiding further diagnostic measures [1].

Pseudogynecomastia, or lipomastia, on the other hand, involves the enlargement of the male breast due to the accumulation of subcutaneous fat tissue [3], which must be distinguished from true gynecomastia [1,3]. Pseudogynecomastia can present either bilaterally or unilaterally; unilateral pseudogynecomastia is particularly rare, with only a few cases documented in the literature [3].

Here, we present the case of a 45-year-old male, employed in a car repair shop, who was admitted to the Surgical Department of the General Hospital of Katerini, Greece, with unilateral enlargement of his left breast (Figure 1a,b). The patient reported that, daily, he put a heavy iron bar under his left axilla and pressed it firmly against his left chest wall during his work at the car repair shop for the last 25 years. He also reported that, during his work at the car repair shop, he was required to work with heavy machinery and sometimes repeated vibrations and microtrauma would affect his left axilla and chest wall. The clinical examination revealed a significantly enlarged left breast, with a horizontal skinfold measuring 40 cm in circumference. The laboratory tests, including serum testosterone, estradiol, luteinizing hormone, and hCG levels, were within the normal limits.

A subsequent mammography and breast ultrasound were performed, revealing lipomatosis with pronounced fibrous tissue but without any enlargement of the breast glandular tissue (Figure 2).

The patient underwent the surgical removal of the excess adipose tissue in the left breast while preserving the nipple–areola complex using the Kornstein technique (Figure 3 and Figure 4a,b). The postoperative course was uneventful, without complications, and he was discharged on the fourth day of hospitalization.

The Kornstein technique involves performing a mastectomy with the repositioning of the nipple [4,5]. In cases of pseudogynecomastia, this technique can be utilized to preserve the nipple and areolar complex by removing the excess fat tissue and repositioning the nipple [4,5].

The histopathological examination revealed lobular adipose tissue measuring 36 × 19 × 14 cm, and it was negative for evidence of a gland mass or malignancy. At the 12-month postoperative follow-up, the clinical evaluation showed a symmetrical appearance of both breasts with no recurrence observed (Figure 5a,b).

A review of the international literature revealed only a few cases of unilateral pseudogynecomastia.

In 2010, Spyropoulou et al. reported a series of five patients who experienced varying degrees of unilateral pseudogynecomastia and shared the commonality of working in the same metal-pressing factory. They hypothesized that pseudogynecomastia might be an occupational risk in such environments due to repetitive trauma caused by continuous vibration and pressure under the axilla and the lateral chest wall, potentially having an anabolic effect on the fatty tissue [6].

In 2004, Arnon et al. reported six cases of unilateral pseudogynecomastia in patients who worked in metal-pressing factories, concluding that trauma could be a causative factor [7].

Erol et al. also presented a case report of a patient with unilateral pseudogynecomastia who sustained chronic vibration, pressure, and irritation to the pectoral region over 25 years of working in a manual metal-pressing factory [8].

In 2020, Hanneken et al. reported a case of unilateral pseudogynecomastia and suggested the existence of a new work-related disorder: mechanically induced unilateral pseudogynecomastia caused by severe fibrosis due to chronic trauma [3].

Kang et al. reported a unique case of gynecomastia in a tennis player in 2012, where chronic trauma led to true gynecomastia rather than just fat tissue growth. They suggested that continuous stimulation could cause local release of growth factors and inflammatory mediators over the anterior chest wall muscles, leading to the differentiation of precursor cells into mature glandular tissue, thus causing gynecomastia [9].

Other cases of unilateral pseudogynecomastia have been reported in the literature, such as breast carcinomas (accounting for less than 1% of cancers in men), neurofibromas, lymphangiomas, hematomas, lipomas, and dermoid cysts, which can often lead to eccentric breast enlargement [10]. These can occur any time in life and should be considered in the differential diagnosis.

In 2011, Durkin et al. reported on a 14-year-old boy who presented with an enlarging unilateral breast mass, which was found to be owing to an intraductal papilloma at the time of surgical excision [11].

In 2017, Pellegrin et al. reported on an 11-year-old boy with a sudden painful enlargement of the left breast as a result of a subareolar hematoma due to trauma from nipple pinching at school [12].

Montero et al., in 2023, also reported on a unique case of a 2-year-old boy with unilateral breast enlargement due to the presence of a galactocele [13].

Here, we report on a case of pseudogynecomastia as a result of occupational trauma. True gynecomastia has also been associated with occupational risks by Laimon et al., who reported on a case of a prepubertal boy with occupational trauma as a cause of true gynecomastia, as confirmed by pathological specimens. They concluded that repeated friction of the breast can lead to true gynecomastia and not only to pseudogynecomastia, as previously believed [14].

Pseudogynecomastia treatment involves the removal of excess fat tissue, which can be achieved by liposuction alone [15]. The surgical techniques typically aim to preserve the nipple–areola complex on a pedicle to maintain neurovascular function, achieve a natural appearance, and retain pigmentation [16]. In cases of significant breast enlargement and extensive resection, free-nipple grafting may also be employed [16]. Another method of treatment for pseudogynecomastia is cryolipolysis, which selectively reduces the breast fat tissue without severe side effects or extended downtime, and it can be used as a noninvasive method [17].

This case report aims to raise awareness among surgeons about the rare occurrence of unilateral pseudogynecomastia as a result of chronic trauma. The inflammation caused by a chronic mechanical injury can lead to the release of growth factors, hematoma degradation products, and inflammatory mediators, resulting in connective tissue fibrosis and pseudogynecomastia [3,6]. Therefore, we suggest that pseudogynecomastia should be considered an occupational risk for individuals working with heavy machinery and experiencing chronic trauma to the pectoral region and lateral chest wall.

## Figures and Tables

**Figure 1 diagnostics-14-02058-f001:**
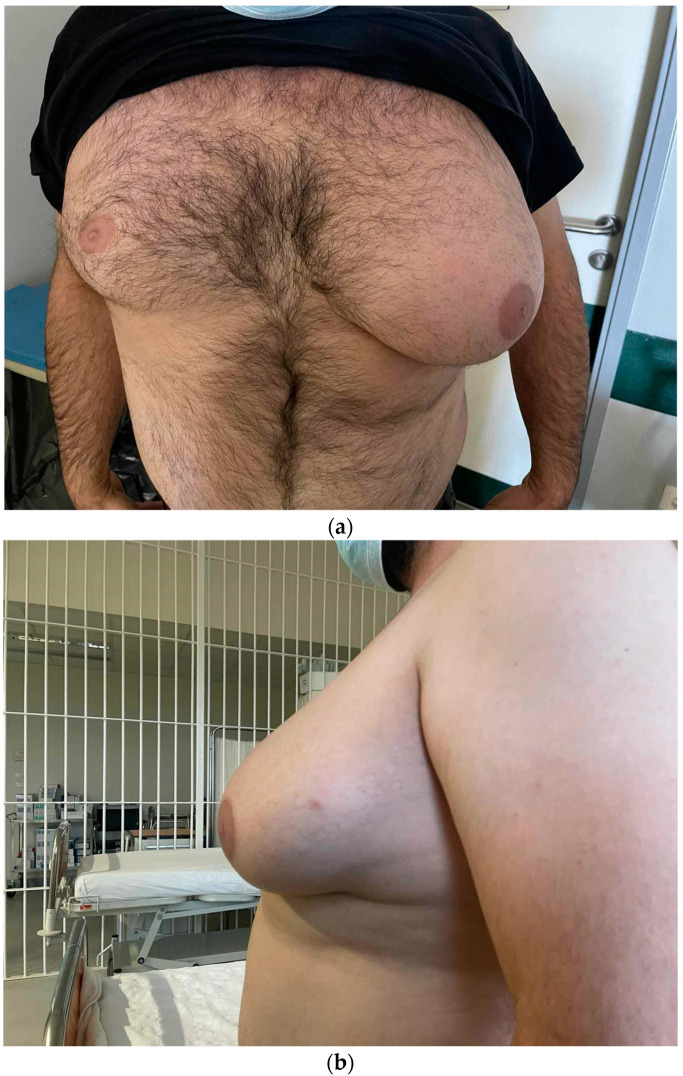
(**a**,**b**) The enlargement of the left breast that extends to the left axilla.

**Figure 2 diagnostics-14-02058-f002:**
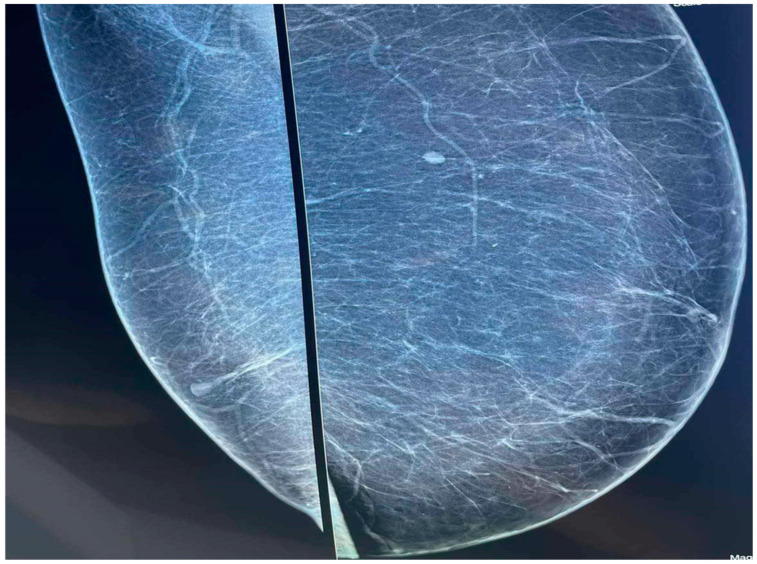
The mammography revealing lipomatosis with pronounced fibrous tissue but without any enlargement of the breast glandular tissue.

**Figure 3 diagnostics-14-02058-f003:**
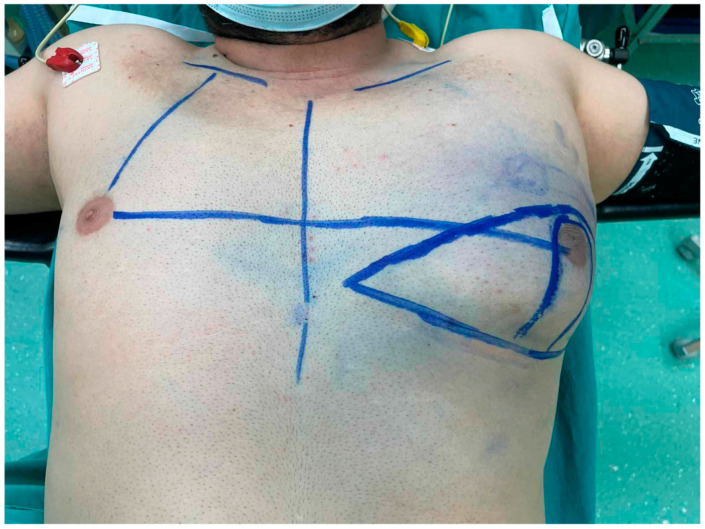
Preoperative markings on the patient’s left breast. The new nipple position was marked to match the approximate position of the contralateral nipple.

**Figure 4 diagnostics-14-02058-f004:**
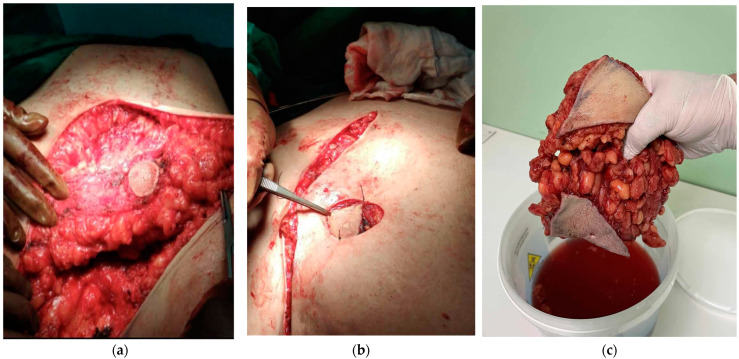
(**a**) The remaining glandular tissue on the left breast after the removal of the excess subcutaneous fat tissue. (**b**) The Kornstein technique with the repositioning of the nipple. (**c**) The specimen that was sent for histopathological examination.

**Figure 5 diagnostics-14-02058-f005:**
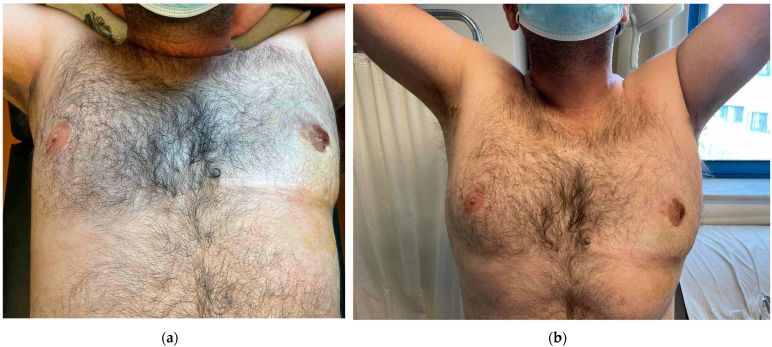
(**a**,**b**) The patient, 12 months after the surgery, experienced partial nipple loss. However, he was satisfied with the final result, and good symmetry was achieved.

## Data Availability

All relevant data are within the manuscript.

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
