# Peer review of "A Unique Case of Unilateral Pseudogynecomastia"

_diagnostics, 2024, doi:10.3390/diagnostics14182058_

Round 1

Reviewer 1 Report

Comments and Suggestions for Authors

The authors present a unique case of unilateral pseudogynecomastia in a 45-year-old male mechanic. The patient exhibited significant enlargement of the left breast, extending to the axilla. Clinical evaluation, imaging, and hormonal assays revealed lipomatosis with pronounced fibrous tissue, but no glandular tissue involvement or hormonal abnormalities. The patient underwent surgical excision of excess adipose tissue using the Kornstein technique, which preserved the nipple-areolar complex. Histopathological examination confirmed the absence of malignancy, and follow-up at 12 months showed symmetrical breast appearance with no recurrence.

Comments

While the patient's occupation is mentioned, details about duration of exposure, specific tasks, and protective measures are not provided. A more comprehensive occupational history would be valuable.

The study doesn't mention using any standardized tools to measure breast volume or patient satisfaction. Incorporating validated assessment tools would improve the objectivity of the findings.

While the authors differentiate between true gynecomastia and pseudogynecomastia, other potential causes of unilateral breast enlargement (e.g., tumors) are not thoroughly discussed. A more comprehensive differential diagnosis would strengthen the paper.

The paper focuses on surgical management but doesn't thoroughly explore non-surgical alternatives. Discussing a range of treatment options would provide a more balanced perspective.

Author Response

Comment 1 : While the patient's occupation is mentioned, details about duration of exposure, specific tasks, and protective measures are not provided. A more comprehensive occupational history would be valuable.

Response 1 : We added : The patient reported that daily he put a heavy iron bar under his left axilla and pressed it firmly against his left chest wall, during his work at the car repair shop for the last 25 years

Comment 2 : The study doesn't mention using any standardized tools to measure breast volume or patient satisfaction. Incorporating validated assessment tools would improve the objectivity of the findings.

Response 2 : There was no need or indication for preop volume assesment. It was and exorbitant case. The objective was to reduce and adjust the enlarged breast to the smaller one.

Comment 3 : While the authors differentiate between true gynecomastia and pseudogynecomastia, other potential causes of unilateral breast enlargement (e.g., tumors) are not thoroughly discussed. A more comprehensive differential diagnosis would strengthen the paper.

Response 3 : We added : Other cases of unilateral pseudogynecomastia have been reported in the literature, such as breast carcinomas (accounting for less than 1% of cancers in men) neurofibromas, lymphangiomas, hematomas, lipomas and dermoid cysts, which can lead often to eccentric breast enlargement.

Comment 4 : The paper focuses on surgical management but doesn't thoroughly explore non-surgical alternatives. Discussing a range of treatment options would provide a more balanced perspective.

Response 4 : As this is a case of pseudogynecomastia , we chose not to thouroughtly discuss the treatment of gynecomastia ( which is a totally different pathology). We addded : Another method of treatment for pseudogynecomastia is cryolipolysis which selectively reduces breast fat tissue without severe side effects or extended downtime, and can be used as a noninvasive method

Reviewer 2 Report

Comments and Suggestions for Authors

I would like to thank Kountouri et al (it is rather awkward that "al" corresponds to 13 authors) for this case report of unilateral pseudogynecomastia. It is an interesting and well presented case. My main comment would be about the lack of (even an attempt to give) an explanation why this patient developed unilateral gynecomastia. The literature presented in the Discussion is more than enough but I would suggest you try to relate this literature with your patient and give a plausible explanation. And a small reference should be also included in your Abstract. 

Either than that, some trivial comments:

Paper's title: erase the full stop at the end. The same at the end of the authors' names.

You should mention somewhere at the early parts of the manuscipt that pseudogynecomasta is also termed lipomastia - which is a more descriptive term about fat acummulation.

Line 34: since Leydig, Sertoli or germ-call tumors can be implicated in gynecomastia ie a palpable mass (further to a possible atrophy of Klinefelter syndrome) you should more precisely describe possible testicular implication in gynecomastia cases.

Line 41: Katerini, Greece

Line 95: Please rephrase to "Pseudogynecomastia treatment"

Author Response

Comment 1: Paper's title: erase the full stop at the end. The same at the end of the authors' names.

Response 1: We will revise.

Comment 2: You should mention somewhere at the early parts of the manuscipt that pseudogynecomasta is also termed lipomastia - which is a more descriptive term about fat acummulation.

Response 2: We added the term

Comment 3: Line 34: since Leydig, Sertoli or germ-call tumors can be implicated in gynecomastia ie a palpable mass (further to a possible atrophy of Klinefelter syndrome) you should more precisely describe possible testicular implication in gynecomastia cases.

Responce : We added :  Approximately 2%–4% of gynecomastia cases are related to testicular tumors, and 7%–11% of patients with testicular tumors present with gynecomastia as the initial symptom 

Comment 4: Line 41: Katerini, Greece

Response 4 : we will revise

Comment 5: Line 95: Please rephrase to "Pseudogynecomastia treatment"

Response 5 : We will revise

Reviewer 3 Report

Comments and Suggestions for Authors

The pathology presented is well described in the preceding literature, and no novel aspects are presented in diagnostic or therapeutic terms.

In the Keywords, an occupational role is attributed to the pathology, but this is not reported in the present case. No reference is made to microtrauma suffered by the patient due to his mechanical condition.

The approach taken was expected to compromise the nipple's vascularization. Why was it not approached with a cranial flap? Why was no plastic surgeon involved in the process? The cosmetic outcome of the nipple-areola complex is not favorable.

No histologic images of the patient are shown.

Comments on the Quality of English Language

Moderate revision needed

Author Response

Comment 1: The pathology presented is well described in the preceding literature, and no novel aspects are presented in diagnostic or therapeutic terms.

Response 1: With our case report we indent to raise vigilance regarding occypational pseudogynecomastia. To our knowledge only a few cases have been reported in the literature.

Comment 2: In the Keywords, an occupational role is attributed to the pathology, but this is not reported in the present case. No reference is made to microtrauma suffered by the patient due to his mechanical condition.

Response 2 : We added: The patient reported that daily he put a heavy iron bar under his left axilla and pressed it firmly against his left chest wall, during his work at the car repair shop for the last 25 years

Comment 3: The approach taken was expected to compromise the nipple's vascularization. Why was it not approached with a cranial flap? Why was no plastic surgeon involved in the process? The cosmetic outcome of the nipple-areola complex is not favorable.

Response 3: The patient was fully satisfied with the post-op result. The surgeon who operated is a certified breast surgeon. Many techniques for the surgical treatment of gynaecomastia have been reported to be effective with reasonable with limited scar formation. Kornstein et al. introduced a technique using an inferior broad but thin de-epithelialised pedicle bearing the NAC and preserving a vital and homogenous base for the superior skin flap. We chose this technique as it generally accepted in the literature. Our personal experience with other patients using this technique had minimal complications. 

Comment 4: No histologic images of the patient are shown.

Response 4: Unfortunatelly no histologic images are unavailable , since the histopathological examination is conducted in a tertiary hospital in a different city.

Round 2

Reviewer 3 Report

Comments and Suggestions for Authors

The manuscript has been considerably improved. Some additional comments:

1.                  I find the occupational aspect of pseudogynecomastia in the present manuscript more exciting and less studied. I think the authors should include it in the title and abstract.

2.                  The physical occupational mechanism potentially associated with this pseudo-gynecomastia should be further explored. Was there repeated microtrauma or repeated vibrations? Was there constant, firm pressure? I recommend expanding.

3.                   I recommend adding a paragraph to the introduction/discussion, briefly stating other rare causes of unilateral breast enlargement in males and clarifying that they can occur at any time of life, even in childhood. Multiple uncommon lesions have been described in unilateral breast enlargement in males that should be considered in the differential diagnosis. Relevant references are attached. 

Arredondo Montero, J., Bronte Anaut, M., Ayuso González, L. et al. Unilateral Galactocele in a 2-Year-Old Boy: the Role of GATA-3. Indian J Surg 85, 1485–1487 (2023). https://doi.org/10.1007/s12262-023-03774-4

Pellegrin MC, Naviglio S, Cattaruzzi E, Barbi E, Ventura A. A Teenager with Sudden Unilateral Breast Enlargement. J Pediatr. 2017 Mar;182:394. doi: 10.1016/j.jpeds.2016.11.044. Epub 2016 Dec 9. PMID: 27956018.

Durkin ET, Warner TF, Nichol PF. Enlarging unilateral breast mass in an adolescent male: an unusual presentation of intraductal papilloma. J Pediatr Surg. 2011 May;46(5):e33-5. doi: 10.1016/j.jpedsurg.2011.02.068. PMID: 21616226.

Similarly, occupational trauma also appears to produce true gynecomastia. Review this reference.

Laimon W, El-Hawary A, Aboelenin H, Elzohiri M, Abdelmaksoud S, Megahed N, Salem N. Prepubertal gynecomastia is not always idiopathic: case series and review of the literature. Eur J Pediatr. 2021 Mar;180(3):977-982. doi: 10.1007/s00431-020-03799-x. Epub 2020 Sep 25. PMID: 32975593.

“…This is the first time to report occupational trauma as a cause of true gynecomastia as confirmed by pathological specimen, in a prepubertal boy…”)

Comments on the Quality of English Language

Minor revision needed. 

Author Response

  1. I find the occupational aspect of pseudogynecomastia in the present manuscript more exciting and less studied. I think the authors should include it in the title and abstract.

Response 1:  We addded The patient reported daily microtrauma on his left axilla and chest wall on the abstract. 

2.The physical occupational mechanism potentially associated with this pseudo-gynecomastia should be further explored. Was there repeated microtrauma or repeated vibrations? Was there constant, firm pressure? I recommend expanding.

Response 2: We added: The patient reported that daily he put a heavy iron bar under his left axilla and pressed it firmly against his left chest wall, during his work at the car repair shop for the last 25 years. He also reported that during his work at the car repair shop, he was required to work with heavy machinery and sometimes repeated vibrations and microtrauma would affect his left axilla and chest wall.  

3. I recommend adding a paragraph to the introduction/discussion, briefly stating other rare causes of unilateral breast enlargement in males and clarifying that they can occur at any time of life, even in childhood. Multiple uncommon lesions have been described in unilateral breast enlargement in males that should be considered in the differential diagnosis. Relevant references are attached. 

Arredondo Montero, J., Bronte Anaut, M., Ayuso González, L. et al. Unilateral Galactocele in a 2-Year-Old Boy: the Role of GATA-3. Indian J Surg 85, 1485–1487 (2023). https://doi.org/10.1007/s12262-023-03774-4

Pellegrin MC, Naviglio S, Cattaruzzi E, Barbi E, Ventura A. A Teenager with Sudden Unilateral Breast Enlargement. J Pediatr. 2017 Mar;182:394. doi: 10.1016/j.jpeds.2016.11.044. Epub 2016 Dec 9. PMID: 27956018.

Durkin ET, Warner TF, Nichol PF. Enlarging unilateral breast mass in an adolescent male: an unusual presentation of intraductal papilloma. J Pediatr Surg. 2011 May;46(5):e33-5. doi: 10.1016/j.jpedsurg.2011.02.068. PMID: 21616226.

Similarly, occupational trauma also appears to produce true gynecomastia. Review this reference.

Response 3: We reviewed all the references and added the information.

Round 3

Reviewer 3 Report

Comments and Suggestions for Authors

There is a minor typo: reference 11 is duplicated (see reference 13), Durkin et al. reference needs to be included, and Montero et al. is referenced as 13 in the main text but appears as no.12 in the references. Please correct during  proofreading and typesetting. 

Comments on the Quality of English Language

Minor polishing needed.